# In Answer to the Pauline Principle: Consent, Logical Constraints, and Free Will

**Marilie Coetsee** 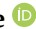

Jepson School of Leadership Studies, University of Richmond, Richmond, VA 23173, USA;
mcoetsee@richmond.edu

**Abstract:** James Sterba uses the Pauline Principle to argue that the occurrence of significant, horrendous evils is logically incompatible with the existence of a good God. The Pauline Principle states that (as a rule) one must never do evil so that good may come from it, and according to Sterba, this principle implies that God may not permit significant evils even if that permission would be necessary to secure other, greater goods. By contrast, I argue that the occurrence of significant evils is logically compatible with the existence of a good God because victims of significant evils may themselves reasonably consent to their suffering. In particular, I argue that they may be able to accept their suffering if it turns out that there was no way for God to secure relevant greater goods (or prevent other, greater evils) except by way of allowing their suffering, and God also provides them with other compensating, heavenly comforts. After using this consent-based argument to address Sterba's *logical* problem from evil, I briefly consider how this argument may also help address a related *evidential* problem from evil, which suggests that while it is possible that victims of significant evils would consent to their suffering, it is *unlikely* that they would do so. While I do not provide a definitive solution to this evidential problem of evil, I highlight one important example of a trade-off that God may need to make that would—along with the provision of compensating, heavenly comforts—potentially persuade victims of significant evils to consent to their suffering. Specifically, I argue that there may be a necessary trade-off that God needs to make between permitting significant evils (on the one hand) and protecting a certain, morally significant form of free will (on the other hand).

**Keywords:** problem of evil; skeptical theism; consent; free will; Pauline Principle; Doctrine of Double Effect

## 1. Introduction

How do we reconcile the existence of a good, all-powerful, and all-knowing God with the far-reaching presence of pain and suffering among God's creatures? It is not difficult to think of reasons why God might allow us to experience *some* pain. Perhaps some measure of pain is necessary for prodding a growth in moral character ([Hick 2016](#)), or perhaps it is a necessary side-effect of God's choice to give us free will ([Plantinga 1977](#)). However, the more one begins to take seriously the scope and intensity of people's suffering, the more difficult it becomes to square that suffering with the supposition that a good, all-powerful God exists. As James [Sterba](#) ([2019](#)) points out, it seems that God could prevent the most significant and horrendous evils while still protecting adequate opportunities for character development and the exercise of free will.[1] For instance, while the freedom of abusive parents might be limited to some extent if God intervened to prevent them from carrying out the full measure of the torment they sought to inflict on their children, such an intervention would by no means need to *wholly* deprive those parents of free will, and the children could no doubt adequately grow their character without having to undergo the agony of abuse. (Indeed, their suffering would surely tend to do more to *undermine*, rather than promote, the healthy development of their character.)

Given these facts about the scale and scope of human suffering, advocates of *evidential arguments from evil* conclude that it is *improbable* that God exists ([Rowe 1996](#)). [Sterba](#) ([2019](#),

2021), however, reaches a stronger conclusion. He argues that the significant, horrendous evils we observe in our world are *logically* incompatible with the existence of a good, all-knowing, and all-powerful God. On his view, it is not just improbable, but *impossible*, that such significant evils could occur in circumstances where a good God also exists.

Sterba's logical argument from evil draws from resources in ethical theory that philosophers of religion have too often overlooked. Many philosophers of religion working on the problem of evil have implicitly presupposed a consequentialist framework (Bergmann 2014; Pike 1963; Plantinga 1977), according to which the 'end' of securing the greatest possible aggregate good can in principle justify the 'means' of permitting some lesser evils (at least assuming that those lesser evils are *necessary* means to securing the greater good). Evidential arguments from evil that draw on this framework may suggest that it is improbable that the scale and scope of suffering we see is necessary to secure any such greater goods. However, Sterba's argument appeals to moral ideas familiar from an alternative, Kantian framework to argue that the scale and scope of suffering we see could not, *even in principle*, be justified just by reference to the (alleged) fact that such suffering may be necessary to promote other, 'greater goods'.[2]

To make his argument, Sterba points readers to the Pauline Principle, which holds that (as a rule) one must never do evil so that good may come from it (Sterba 2019, Chapters 1 and 4). *Even if* the horrendous evils we observe were necessary means to achieving greater goods, this principle implies that God's greater 'ends' still could not justify the 'means' of God's permitting the (supposedly) 'lesser' evils we observe. Even if letting a child be tormented by their abusive parents was offset by some 'greater good', for instance, God would (on Sterba's view) still *in principle* be wrong to allow it. Thus, the occurrence of horrendous evils not only seems to make the existence of a good God *unlikely* but (onSterba's view) altogether *rule out* the possibility of the existence of such a God.

As Sterba acknowledges, the Pauline Principle admits of some exceptions. It may, for instance, be permissible to use lesser evils as means to securing greater goods if those evils are trivial or easily reparable (Sterba 2019, pp. 2–3*ff* and 49–50*ff*, 76). Sterba also grants that allowing or perpetuating even significant harm may be justified if that allowance is the only way to prevent even greater harm from befalling other innocent victims. For example, if the only way to prevent a military despot from killing twenty innocent civilians is to kill one innocent civilian yourself, you may potentially be justified in doing so. In *Is a Good God Logically Possible?*, Sterba addresses such exceptions to the Pauline Principle up front, and he focuses considerable effort on arguing that they would not suffice to excuse God in permitting horrendous evils (ibid.).

However, there is also one additional route by which exceptions to the Pauline Principle may be justified, which Sterba allots less attention to: namely, lesser evils may be permitted in pursuit of greater goods if those who undergo those lesser evils consent to their suffering.[3] In this paper, I use this kind of argument—which I call the *Consent Argument*—to argue against Sterba's logical argument from evil. I argue that the existence of significant evils is not *logically* incompatible with the existence of a good (all-powerful, all-knowing) God because victims of significant evils may themselves reasonably consent to their suffering.

My argument proceeds as follows. In Section 2, I outline the Consent Argument and address Sterba's initial objections to it. In Section 3, I draw on the resources of skeptical theism to argue that God's allowance of significant evils may, for all we know, be logically necessary to securing greater goods or preventing even greater evils. If victims of significant evils are provided with compensating, heavenly comforts *and* there does turn out to be this kind of logical trade-off between their earthly suffering and the realization of other, greater goods (or the prevention of other, greater evils), then, I suggest, even victims of significant evils may in principle consent to their suffering.

I take it that the arguments of Sections 1 and 2 suffice to show that Sterba's *logical* argument from evil does not stand, at least as it is currently articulated. For if, for all we know, there are grounds on which victims of significant evils can themselves reasonably

consent to their suffering, then we are not yet in a position to conclude that those evils are logically incompatible with the existence of a good God. Still, this response to Sterba's logical argument from evil does not provide a positive explanation of why God may need to permit significant evils or specify the grounds on which victims may consent to those evils. As I discuss in Section 4, without these further details, the evidential problem of evil still persists. In Sections 4 and 5, I begin to address the evidential problem of evil by exploring one candidate explanation for why victims of significant evils may reasonably consent to their suffering. In particular, I contend that victims of significant evils may consent to their suffering because of certain trade-offs that exist between God's making room for such suffering and the preservation of a morally significant form of free will.

## 2. The Consent Argument

In the course of addressing skeptical theist responses to his argument, Sterba (2019) entertains the possibility that God's choice to permit innocent individuals to undergo significant suffering may be excused if those individuals themselves gave their informed consent to such suffering. As he notes, our earthly observations suggest that "nothing like informed consent typically obtains" (Sterba 2019, p. 74). Sterba then also briefly considers the possibility that victims may consent *retroactively* to their earthly suffering in a heavenly afterlife. While the claim that one may be able to retroactively consent to decisions that impact you (like God's decision to allow your earthly suffering) might initially seem strange, Sterba's use of this language in his argument is plausibly meant to convey the idea—commonly agreed to by authors writing on contractualist ethics and political philosophy—that decisions can be morally justified by the fact that they are *or would be* reasonably acceptable to those impacted, given adequate information.[4] I will follow Sterba in speaking interchangeably about retroactive consent and 'acceptability' and assume with him (and others) that retroactive consent can at least sometimes have normative power.[5] Nevertheless, Sterba goes on to argue that retroactive consent may not be forthcoming in the case of those who have suffered significant evils because God's permission of the relevant suffering seems to violate the Pauline Principle. Given that God ought not to have allowed evil in the interest of securing greater goods, Sterba writes, "victims may never be able to... find reasonably acceptable the infliction of such [suffering] on themselves" (Sterba 2019, p. 75). Sterba presupposes that God's permission of suffering violates the Pauline Principle and uses that as a basis to argue that victims of significant evils would not consent to their suffering. But Sterba is not entitled to assume that God's permission of suffering violates the Pauline Principle *unless* he can already show that the relevant consent would not be forthcoming. As Sterba notes, the Pauline Principle sits at the heart of the Doctrine of Double Effect, and—as Warren Quinn argues—this doctrine plausibly reflects Kantian ideals. As Quinn points out, it is plausibly the Kantians requirement that we show respect for others as ends-in-themselves—that is, that we show regard for them as rational autonomous agents with inviolable dignity and worth—that explains why individuals have a right (as Quinn puts it) "not to be sacrificed in strategic roles over which they have no say" or "to be pressed . . . into the service of other's people's purposes" (Quinn 1989)—even if those purposes serve some other, greater good. But while Kantian ideals explain why it would, *in general*, be wrong for God to violate the Pauline Principle, those ideals also suggest that the *consent* of individuals to undergo relevant instances of suffering would excuse God's choice to permit those sufferings. For if individuals, by their own rational, autonomous choice accept their suffering, then God would not show disregard for their rights by permitting them to suffer as a means to promoting a greater good (or preventing an even greater evil). Call this the *Consent Objection* to Sterba's logical argument from evil.

In *Horrendous Evils and the Goodness of God*, Marilyn McCord-Adams (2000) provides resources for developing at least one version of the *Consent Objection*. There, she suggests that God's permission of horrendous evils may be excused at least in part by virtue of the (supposed) fact that God could guarantee to those who suffer horrors a life which is, on the whole, a great good to them—for instance, by offering them an experience of Divine beauty

and goodness in an afterlife that is "immeasurable and incommensurate with . . . created goods or ills" ([McCord-Adams 2000], p. 147). McCord-Adams notes that such beauty and goodness must be "great enough" to "defeat"—or, we might add, in some other sense compensate for—relevant sufferings not only from an external point of view, like God's, but also *from the point of view of the person* receiving those goods (ibid., p. 145). If victims of horrendous evils can, in the afterlife, enjoy goods that they themselves recognize to be "immeasurab[ly]" better than the suffering they experienced on earth, and they accept that this makes their life on the whole a great good to them, that may (it seems) open up the possibility that they would retrospectively accept their worldly suffering.

Such heavenly acceptance of this-worldly suffering, however, cannot simply be taken for granted. In particular, one cannot reasonably expect victims to consent to their suffering if there turns out not to have been any good reason for them to undergo it. Consider an analogy. A grown-up child might reasonably continue to object to the inaction of a parent who, for little reason or simply out of indifference, stood by when she was undergoing harrowing pain—*even if* that parent later tries to curry favor with her by inviting her to live with him rent-free in his mansion in Los Angeles. Indeed, that child may not only *not* consent to her prior suffering but also reasonably *reject* her parent's invitation to the mansion. No matter how 'heavenly' the material conditions of life in the mansion, the prior indifference the parent showed toward her suffering might make unpleasant the thought of continued life with him. Similarly, victims of significant evils may not consent to their suffering—*even if* God provides great heavenly comforts—if God simply stood by and watched their suffering when their suffering was not necessary to secure any other important values.

The success of the *Consent Objection* thus depends not only on the future provision of sufficient heavenly comforts, but also on the past necessity of God's having *needed* to make certain, reasonable trade-offs to secure other moral goods that required relevant victims to suffer. As noted in the introduction, Sterba grants that, in theory, God's permitting significant evils could be excused *if* that permission were genuinely necessary to avert an even greater evil. In making that admission, Sterba did not take into account the possibility that victims might consent to their suffering. Once we take this possibility of consent into account, we might plausibly *also* argue that God's permission of significant evils could be excused if that permission where genuinely necessary to secure a greater good. Specifically, if victims themselves found that greater good to merit their willing, sacrificial suffering and—in combination with their own later enjoyment of heavenly comforts—thus found there to be sufficient reason to consent to that suffering, then (it seems) God's 'trading-off' their suffering for the sake of securing some other, greater good could also be excused. I will speak interchangeably from this point on of trade-offs meant to secure greater goods and trade-offs meant to prevent greater evils. I will call the argument that victims of significant evils might possibly consent to those evils, given (i) their future enjoyment of incomparably wonderful heavenly goods and (ii) the this-worldly trade-offs that must be made between those evils and other even greater evils *or* other, greater goods, the *Consent Given Trade-Offs* argument.

If the *Consent Given Trade-Offs* argument succeeds, Sterba's logical argument from evil fails. For, if it is possible that victims of significant evils might consent to their suffering, then significant evils are logically compatible with the existence of a good (all-powerful and all-knowing) God. However, Sterba might object to *Consent Given Trade-Offs* on the grounds that there are no relevant trade-offs that an *omnipotent* God would be required to make between (on the one hand) preventing significant evils and (on the other hand) preventing other even greater evils (or securing other even greater goods). I address this objection in the next section.

## 3. Skeptical Theism and Divine Trade-Offs

While Sterba's commitment to the Pauline Principle keeps him from giving substantial consideration to the possibility that God may be justified in permitting significant evils in

order to secure greater goods, he gives more serious attention to the possibility that God may be excused in permitting significant evils if that permission is necessary to prevent an *even greater evil*. As he points out, if the only way to prevent a military despot from shooting twenty innocent civilians is to do as she asks and shoot one innocent civilian yourself, you may be justified in doing so. We might be inclined to think that the same kind of excusing argument for shooting the innocent civilian that applies to you in this case could be extended to apply to God's choice to permit significant evils.

However, Sterba ultimately denies that this extension can be made; God's permission of lesser evils, he argues, cannot be justified by reference to a supposed need to prevent even greater evils. As Sterba argues, this is because God would not face the kinds of limitations that we do in having to make trade-offs between allowing lesser evils and preventing more significant evils. Whereas we lack the causal power to simultaneously save the one and the twenty, an all-powerful God could (Sterba points out) do both (Sterba 2019, p. 50). God could, for instance, refuse to kill the one and then cause the military despot's guns to malfunction or misfire to also prevent her from shooting the other twenty civilians. Although Sterba does not discuss this argument as it applies to trade-offs made for greater goods, we can assume that he may also try to apply there; Sterba may argue that because God is all-powerful, God would not be constrained to make trade-offs between allowing significant evils and securing greater goods. Call this objection to my *Consent Given Trade-Offs* argument the *No Trade-Offs* objection.

In defending the *No Trade-Offs* objection, Sterba makes reference to cases—like the despot case noted above—where God's use of God's unlimited *causal* powers is what seems to make it possible for God to both prevent a significant evil *e* and *simultaneously* prevent a greater evil *E*. But Sterba's defense presupposes that the relevant trade-offs between *e* and *E* are always of a causal nature; that is, Sterba assumes that absent divine intervention, if a significant evil *e* is prevented, that would (only) *causally* necessitate the occurrence of the greater evil *E*. Scott Coley (2021) challenges this assumption. Drawing on the resources of skeptical theism, Coley suggests that (for all we know) there may be *logical* entailment relations between such *e* and *E*: it may be that if one prevents *e*, that would then *logically* entail that *E* must occur (Coley 2021). For all we know, there may also be logical entailment relations between significant evils *e* and much greater goods *G*, such that if one prevents *e*, that would logically entail that *G* could not be realized. If either kind of logical entailment relation exists (between *e* and *E* or between *e* and *G*), then even an all-powerful God would face some trade-offs in their decision to prevent significant evils. God would no more be able to prevent relevant significant evils *e while simultaneously preventing E* (and/or securing *G*) than God would be able to create a round square. Call this response to the *No Trade-Offs* objection the *Logical Trade-Offs* response.

Coley grants that the notion that there may be *logical* entailment relations between the occurrence of significant evils and prevention of even greater evils (or the securing of even greater goods) may seem "truly foreign to us." (ibid., p. 2). Most trade-offs between significant evils and greater goods (or greater evils) that *we* are familiar with may well be of the causal kind that Sterba has in mind, and so it is not surprising that Sterba's *No Trade-Offs* argument focuses on such cases. But, as Coley rightly points out, the fact that we have difficulty imagining a logical entailment relation between a significant evil *e* and a greater good *G* (or greater evil *E*) does not show that such entailment relations do not exist. Indeed, as Coley points out, *skeptical theists* might well argue that it is unsurprising that such entailment relations are not immediately imaginable to us since (skeptical theists might say) there is little reason to expect that the entailment relations that *we* are familiar with are representative of the kind of entailment relations there actually are (ibid.).

Sterba's response to Coley's argument takes a strange turn, and I will only briefly address it here. At one point in his argument, Coley says that "in terms of causal powers, God is more powerful than we are" (ibid.). Coley says this with the apparent aim to suggest that, even if God is logically prevented from *simultaneously* preventing *e and* preventing *E* (or securing *G*), God may still be more causally powerful than us in having the capacity to

*either* prevent *e*, taken in isolation, *or* prevent *E* (or secure *G*), taken in isolation. However, Sterba interprets Coley as making the claim that God is more powerful than us because God has a causal ability to prevent each significant evil *e* while also simultaneously preventing greater evils *E* (or securing greater goods *G*). Given this reading of Coley's argument, Sterba understandably objects: "Coley's argument fails," he writes, "because neither God nor anyone else could be causally able to do what is logically impossible for them" (Sterba 2021, p. 20). Coley's argument, he concludes, "is based on the possibility of an impossibility and so does not work." (ibid.)

To illustrate the structure of Coley's argument and help respond to Sterba's objection, it may be useful to consider an example. Consider *Professor's Dilemma:*

> *Professor's Dilemma:* Professor Deos has unlimited causal power to give her student Morty any grade she wishes. Suppose that there is something intrinsically bad about giving a student a failing grade, but that Morty's work also clearly does not merit anything better. Suppose also that considerations related to fairness would make it even worse for Professor Deos to inflate grades and so give Morty a passing grade.

In this scenario, Professor Deos has the causal power to prevent Morty from suffering the evil of failing (*f*) and also has an independent causal power to prevent the even greater evil of grade inflation (*I*). Professor Deos thus is more powerful than Morty, who can do neither *f* nor *I*. Still, even though Professor Deos has causal power to prevent *one* of either *f* or *I*, the logical entailment relations between *f* and *I* still constrain her from preventing *both f* and *I*. For, given the nature of Morty's work and the defined standards of earning a passing grade in the class (we may suppose) there is no logically coherent way for Morty both to get a passing grade *and* to avoid grade inflation. Even though Professor Deos is causally omnipotent with respect to assigning grades, she is also still logically constrained to permit Morty to suffer the lesser harm of getting a failing grade if she is going to prevent the even greater evil of grade inflation.

Like Professor Deos, God may be more casually powerful than us in being able to prevent either *one* of a significant evil *e* or a greater evil *E*, while nevertheless *lacking* the causal power to simultaneously prevent *e and* prevent *E*—precisely because it is logically impossible to prevent *e* while also preventing *E*. If this is right, then God may face genuine trade-offs between preventing significant evils and preventing even greater evils; the same logic suggests that God may face genuine trade-offs between preventing significant evils and securing even greater goods. What's more, if God does face such trade-offs, then victims of significant evils may come to regard God's choice to allow significant evils as reasonable, and (so) consent to their suffering—at least assuming that they are also offered heavenly comforts which make their own lives, on the whole, a great gift to them. If, as I have argued, this *Consent Given Trade-Offs* argument is right, then Sterba's logical argument from evil does not succeed (at least not as it is currently articulated). For if it is logically possible that victims of significant evils can themselves reasonably consent to their suffering, then it is not logically impossible that such evils could co-exist with the presence of a good, all-powerful, and all-knowing God.

## 4. A Logical Entailment between Permitting Significant Evils and Protecting Significant, Free and Effective Choice

Even if, as I have argued, the *Consent Given Trade-Offs* argument shows that Sterba's logical argument from evil fails, an *evidential* argument from evil may still succeed. Evidential arguments from evil often presuppose a consequentialist framework, but Sterba's work provides the basis for a novel kind of evidential argument based on *deontic* premises. In particular, one might draw on Sterba's work to argue that the occurrence of significant evils serves as good evidence against the existence of a good God because it is very unlikely that there are logical entailment relations between allowing significant evils and preventing other greater evils (or securing other greater goods); this (one might argue) makes it very *unlikely* that relevant victims of significant evils would consent to their suffering.

In this section and the next, I address this deontic form of an evidential argument from evil. Although I do not pretend to neutralize the argument, I aim to reduce its persuasive force by discussing at least one concrete example of a case where God may be logically constrained to make a trade-off between significant evils and greater goods and where victims of significant evils might regard that trade-off that God must make as a reasonable basis for consenting to their suffering. Specifically, I argue that God may need to make a trade-off between preventing significant evils on the one hand, and, on the other hand, giving human beings a capacity for a certain kind of morally significant free will—a form of free will that I refer to as *significant, free and effective choice*. I suggest that there is a logical, conceptual connection between possessing this kind of free will (on the one hand) and (on the other hand) having the ability to carry out significant evils, and that as a result, God could only avoid the occurrence of significant evils if God gave up on securing the alternative good of significant, free and effective choice. In the next section, I then return to discuss when and why victims of significant evils may consent to their suffering given their knowledge of this trade-off between protecting significant, free, and effective choice and permitting significant evils that God must make (and assuming an additional, appropriate provision of compensating heavenly comforts).

According to Sterba, God does not need to make any trade-off between preserving morally significant forms of free will and allowing significant evils. In particular, Sterba contends that God could prevent significant evils while simultaneously preserving for human beings a morally significant form of free will by simply preventing each perpetrator of a significant evil from successfully completing the *final step* of her action, with its horrible consequences for victims (Sterba 2019, p. 21). Sterba grants that, by intervening in the last moment of (say) an assaulter's attempt to assault her victim, God would restrict some aspect of her freedom to 'successfully' inflict harm on her victim. However, he argues that the protection of *this* aspect of her freedom should not take a moral priority over the freedom of the relevant victim *not* to be harmed (ibid., Chapters 2 and 4). The loss of external efficacy of the assailant's choice to inflict significant harm is thus (according to Sterba) not morally worrisome or lamentable.

Sterba suggests that God can intervene to prevent the final steps of someone's act to commit a significant evil and still allow her full freedom in planning, intending, and "even tak[ing] initial steps toward carrying out" her immoral actions (ibid., pp. 51, 53). By contrast with the loss of freedom to effectively inflict pain—which Sterba takes *not* to be morally significant—Sterba seems to acknowledge that there is something morally valuable in protecting this form of 'inner,' psychological freedom, i.e., someone's freedom to consider different possible good and evil ends, evaluate their merits, and form and act on intentions to pursue those ends. Thus, Sterba denies that God's decision to block the final steps of acts of significant evil would reduce our freedom to a kind of 'kindergarten' or 'playpen' freedom in part on the grounds that that blockage would not interfere with anyone's use of these relevant 'internal' freedoms to plan, intend and act on the intention to commit a significant evil (ibid., pp. 53–54). For now, I follow Sterba in taking the moral significance of such inner freedoms for granted; I will discuss them in more detail in the next section. Since Sterba's argument claims that God can block the external consequences of choices to commit significant evils without obstructing any morally significant exercise of free, inner agency, I will call Sterba's argument the *Unobstructive External Intervention* argument—or simply the *Unobstructive Intervention* argument.

The *Unobstructive Intervention* argument is particularly credible when considered against the backdrop of the larger, Kantian framework Sterba presupposes. On standard interpretations of this framework, all the elements of choice that matter for the evaluation of someone's moral agency are located *within* the psychology of the agent herself. Thus, all the *morally significant* aspects of a villain's choice to commit a significant evil remain intact so long as she considers the reasons for pursuing her ends, freely forms an intention to commit a significant evil in pursuit of those ends, and then freely acts on that intention. If, by bad luck, her action does not bring about the external *effects* she intended it to, that need

not (in a Kantian framework) necessarily undermine the freedom of any *morally significant* aspect of her choice.

One intuitive objection to Sterba's *Unobstructive Intervention* argument targets the Kantian approach to moral agency that that argument seems to presuppose. This objection insists that a morally significant form of freedom requires not only 'inner' abilities to choose and act on intentions but also 'outer' abilities to—as Michael Murray puts it—"affect the course of the world" by way of one's actions (Murray 2008, p. 136). I will follow Murray in saying that morally significant forms of agency thus require not only (inner) *free choice* but also (outer) *effective choice* (ibid.). Murray (2008) and Hasker (2020) use this kind of argument about outer effective choice to suggest that Sterba reduces human freedom to a kind of 'kindergarten' or 'playpen' freedom. However, as alluded to above, Sterba has addressed this argument: he claims that there is no significant moral loss to depriving human beings of effective choice when it comes to significant evils because those evils interfere with the more important right that victims have to be free from being victimized. Sterba also notes that, even if God prevented human agents from exercising effective choice when it came to choices to inflict *significant harm* to others, God could still preserve their capacity for effective choice when it came to many other wrong and moderately hurtful choices (Sterba 2021).

More may be said on behalf of Murry and Hasker's arguments about effective choice. For instance, if victims of significant evils themselves thought that giving human beings a capacity for effective choice when it comes to significant evils was 'worth' the costs to their own earthly freedom, and so consented to God's making that trade-off, then Sterba's argument on behalf of the priority of victims' freedom would not succeed. However, rather than focusing just on arguments about the value of (outer) effective choice, per se, I will proceed to highlight the relationship *between* (outer) effective choice and (inner) free choice.

*Even if* one grants that the most morally significant aspects of an exercise of free choice reside within the psychology of the chooser, I contend that 'external' facts about what it is (or is not) possible for you to efficaciously carry out may still matter because those 'external' facts can constrain the scope of the kind of 'inner' moral agency that is available to you. Against Sterba's *Unobstructive Intervention* argument, I will argue that interventions that God makes with respect to the external consequences of agents' choices will thus *also* restrict what kind of exercise of 'inner' moral agency is available to them with respect to those choices. If my argument is right, then God may face a trade-off not only between allowing significant evils and preserving efficacious choice (as Murray and Hasker suggest) but also between allowing significant evils and preserving a certain kind of robust, *inner* moral agency—a kind of agency that Sterba himself seems to acknowledge is important for raising us up and out of the sphere of 'kindergarten' or 'playpen' freedom. There is a trade-off, in other words, between preventing significant evils (on the one hand) and (on the other hand) protecting significant, (inwardly) free and (externally) efficacious choice.

Briefly put, I contend that if God makes it impossible for an agent to effectively carry out a particular intention to $\varphi$, then that impossibility can undermine her capacity to even coherently consider and act on an intention to $\varphi$. Below, I consider two versions of this response to *Unobstructive Intervention*. First, I put forward what I call the *Impossible Intentions Objection*. According to this objection, one can only make an *intentional* choice to $\varphi$ if one has certain *beliefs* about the possibility of successfully $\varphi$-ing, and God's consistent intervention to prevent significant evils from successfully being brought about would thus end up undermining agents' capacity to coherently form and act on intentions to commit significant evils. Second, I advance (what I call) the *Impossible Alternatives Objection.* I suggest that if, as Sterba contends, God's good nature logically required God to always prevent people from carrying out significant evils, and so made the existence of such evils impossible, *that* would also make it impossible for human agents to even coherently entertain *carrying out a significant evil* ($\varphi_e$-ing) as a possible alternative to pursue in action.

Michael Murray alludes to, but does not fully spell out, the *Impossible Intentions Objection*. In *Nature Red in Tooth and Claw: Theism and the Problem of Animal Suffering*,

Murray (2008) tells the story of himself as a young schoolboy who, together with a group of other young kindergartners, set out to fly off a concrete wall in their playground. Murray recounts that, after about twenty minutes of experiment—in which a variety of children with different degrees of strength and skill all found themselves continuously crashing to the ground—they all decided that flying was simply "not in [their] future." (Murray 2008), p. 137. Having decided this, he notes, neither he nor (as far as he knows) any of his kindergarten peers ever made further attempts at self-propelled flight. Moreover (he writes), he suspects that "none of my kindergarten companions *could* now even form the intention to fly off the wall." (ibid.)

Murray does not go beyond reporting his 'suspicion' that he and his friends *could* no longer coherently form the intention to fly, but that suspicion would gain wide support from philosophers of action, many of whom take there to be a close conceptual connection between intending to $\varphi$ and having certain beliefs about the possibility of *successfully $\varphi$-ing*. To begin, consider the 'strong' cognitivist claim that *intending to $\varphi$* just is a matter of believing that one will $\varphi$ and do so precisely because of that intention.[6] Although space does not allow for a full review of the standard arguments for this claim, one can begin to appreciate the motivations for it by noting, as Paul Grice (1971) did, that there seems to be something odd, and even paradoxical, about asserting that "I intend to do $\varphi$, but I might not do it." Strong cognitivists hold that the unintelligibility of such claims shows that having an *intention* to $\varphi$ conceptually requires one to also possess the *belief* that one *will $\varphi$*.

If the strong cognitivist understanding of intention is right and if, as Sterba suggests, a good God would always prevent people from effectively bringing about significant evils, then (it seems) it would not take long before human beings would no longer be able to coherently form intentions to carry out such evils. Just like it did not take long for Murray and his kindergartner friends to realize that their attempts to fly would inevitably fail, it would also not take long for human beings to realize that their attempts to commit significant evils would inevitably fail. Since human beings would no longer be in a position to *believe* that they could successfully carry out significant evils, they would also no longer be in a position to coherently form or act on intentions to carry such evils. In this way, the external limits God placed on efficacious action would constrict the scope of our inner agential capacity to choose.

Sterba responds to Murray's version of the *Impossible Intentions Objection* by pointing out that a wrongdoer could doubt her capacity to $\varphi_e$—that is, carry out a significant evil *e*—while still anticipating that she *would* be able to $\varphi_e{}^*$—that is, carry out some close variant of *e*, *e\**. For instance, even if a wrongdoer came to realize that she could not successfully carry out an attempt to torture someone by waterboarding them, she might nevertheless still believe that she would be able to torture that person by subjecting him to extreme sleep deprivation. At first glance, Sterba's response seems on track. Granted, it does seem that past failures to commit significant evils would eventually *detract* from the strength of such a wrongdoer's belief that she would be successful in bringing about the *next* evil *e\** she tried to commit. However, one might argue on Sterba's behalf that the wrongdoer would not need to have certainty about the success of $\varphi_e{}^*$-ing to coherently intend to $\varphi_e{}^*$. As Robert Audi (1973) suggests, it's quite plausible that I could (for instance) intend to go see a friend for the weekend even if I am not *certain* that (say) my flights will not be cancelled (Audi 1973). Sterba might thus insist that a wrongdoer could form and act on the intention to *carry out a novel significant evil* ($\varphi_e{}^*$) even if God had prevented other, past instances of attempts at significant evils from being successful.

In line with the above argument, I agree that a wrongdoer would not need to believe that she would *definitely* be successful in $\varphi_e{}^*$-ing in order to coherently intend to $\varphi_e{}^*$. However, I contend that someone contemplating $\varphi_e{}^*$-ing would still need to believe that she could at least *probably* $\varphi_e{}^*$. As Audi points out, even if an intention to go visit my friend's house is compatible with an acknowledgement that it is *possible* that I will not successfully arrive there, it would surely still be incoherent for me to tell my friend that "I intend to come visit you for the weekend, though I believe it *improbable* that I will do so."[7]

As human beings continued to try to commit significant evils, they would over time come to realize that a wider and wider variety of attempts at such evils ($\varphi_e \ldots \varphi_e*$) had all come to naught. So, over time, it would become increasingly obvious that one would not be able to successfully carry out the next, new kind of significant evil $\varphi_e*_{+1}$; eventually we would consider it at least improbable, if not nearly impossible, that $\varphi_e*_{+1}$ would succeed. God's consistent interventions to prevent acts of significant evil from being successful would thus still, in the long run, undermine human beings' capacity to coherently form and act on intentions to bring about such evils.[8]

If *Impossible Intentions* is right, then God could not prevent significant evils from being successfully carried out without thereby also imposing corresponding limits on human beings' inner freedom to select and act on intentions to commit such evils. In the next section, I consider in more detail whether and when victims of significant evils might treat this trade-off between allowing significant evils and preserving the relevant kind of inner free choice as part of an acceptable justification for God's having permitted those evils to occur. First, though, I complete this section by briefly exploring another way in which God's intervention to prevent significant evils might serve to restrict the scope of human beings' inner, free moral agency.

Suppose that Sterba is right, and that if a good, all-powerful, and all-knowing God exists, God's nature would *logically* require God to prevent anyone from ever being able to effectively carry out a significant evil. Suppose further that, as many classical theists hold, if such a God exists, then that God also exists *necessarily*. In particular, suppose that—if there is such a God at all—then there is no possible world in which that God could *not* exist.[9] If one accepts both of these claims, then if God exists, it also follows that there could be no possible world in which significant evils could be successfully carried out. The act of *successfully carrying out a significant evil* ($\varphi_e$-ing) would not only be outside of our *causal* reach; $\varphi_e$-ing would also be metaphysically impossible. Call this claim about the metaphysical impossibility of $\varphi_e$-ing the *Impossible Evils* claim.

If *Impossible Evils* is true, then—at least on some standard theories of representational content—it would also be impossible for us to have coherent thoughts including the *content* '$\varphi_e$-ing'. According to advocates of possible world semantics, for instance, the meaning of a proposition $p$ is constituted by the set of possible worlds in which $p$ is true.[10] Since any proposition $p_{\varphi e}$ that involved '$\varphi_e$-ing' would, in Sterba's proposed scenario, not have *any* possible worlds in which it would be true, there would be nothing for a thought pertaining to '$\varphi_e$-ing' to refer to; thoughts about the possibility of '$\varphi_e$-ing' would lack any substantial content or meaning. Causal theories of mental representation deliver a similar verdict. Very roughly, these theories hold the content of a mental representation $m$ is determined by the object or state of affairs that does—or at least would under idealized conditions—reliably cause $m$ to occur.[11] As Roy Sorensen (2002) points out, if there is no possible objects, states of affairs, or conditions that could causally trigger the formation a particular kind of mental representation $m$, then $m$ would (on a causal theory of representation) lack any genuine content (Sorensen 2002). Thus, one might plausibly conclude that on a causal theory of mental representation, if '$\varphi_e$-ing' was metaphysically impossible, there could be no mental representation with the content '$\varphi_e$-ing'.

If either of the two above prominent theories of representational content are right and *Impossible Evils* is also true, then—in a scenario where God exists and prevents all significant evils—human agents would not only be unable to coherently intend to '$\varphi_e$' but also be unable to coherently *conceive* of the possibility of $\varphi_e$-ing, coherently imagine $\varphi_e$-ing, or coherently consider the moral merits or demerits $\varphi_e$-ing. Trying to imagine, consider, or evaluate the act of $\varphi_e$-ing would be on a par with trying to imagine, consider, or evaluate the act of ($\varphi_C$) *putting colorless green ideas to sleep*, or ($\varphi_{XYZ}$) *ingesting XYZ, rather than H$_2$O, by way of drinking water*. Since $\varphi_e$-ing would be equally as impossible as $\varphi_{C\text{-}ing}$ or $\varphi_{XYZ}$-ing, the thoughts of a person who considers the 'possibility' of $\varphi_e$-ing would be equally confused and meaningless as the thoughts of a person who considered the 'possibility' of $\varphi_C$-ing or $\varphi_{XYZ}$-ing. In a scenario where God prevented all significant evils, and was, along the lines

of Sterba's suggestion, *logically* constrained to do so, there would simply be no coherent alternative of '$\varphi_e$-ing' for moral agents to even consider or think about in the first place. Call this the *Impossible Alternatives Objection*.

The *Impossible Alternatives Objection*, like the *Impossible Intentions Objection*, suggests that God's interference with the external consequences of actions that would otherwise give rise to significant evils ($\varphi_e$) has implications not just for the scope of (outer) effective choices to $\varphi_e$ but also for the scope of our capacity for *inner* free choices to $\varphi_e$. Specifically, God's consistent interference with the external consequences of $\varphi_e$-ing would make it impossible for us to meaningfully consider $\varphi_e$-ing and/or form or adopt an intention to $\varphi_e$, and so we could no longer meaningfully *choose between* committing a significant evil or not. We might be able to choose between committing a moderate evil or not or acting to bring about a significant good or not, but a coherent thought of committing a significant evil or not either could not occur to us, or even if it could, it could not coherently be translated into any meaningful intention to act on it.

## 5. Consenting to God's Trade-Offs: The Goodwill and Significant, Free and Effective Choice

In Section 2, I argued that the existence of a good God would be logically compatible with the occurrence of significant evils if the victims of those evils gave their informed consent to their suffering. In Section 3, I suggested more specifically that such informed consent might be forthcoming if the suffering of those victims was in some sense compensated for by their eternal enjoyment of heavenly comforts, *and* they could see that God's permission of significant evils was necessary to prevent even greater evils or secure even greater goods. Finally, in Section 4, I gave one example of a case where a permission for significant evil might be necessary to secure a potentially greater good, that is, the good of human's capacity for significant, free and effective choice.

Will victims of significant evils consent to God's making a trade-off in favor of preserving significant, free and effective choice at the cost of their suffering? It is difficult to say. As I've noted, the *possibility* that they might suffices to address Sterba's logical argument from evil. However, to address an evidential argument from evil, one must show not only that it is possible that victims of significant evils would reasonably consent to their suffering but also that they would *likely* do so. I do not establish this likelihood here and so do not pretend to resolve the evidential argument from evil. Still, I highlight the value of significant, free and effective choice with an aim to better elucidate why victims of significant evils might potentially regard that good as a reasonable ground on which God might have allowed them to suffer.

First, consider some of the costs of a 'playpen' freedom in which individuals are unable to make *efficacious* choices to commit significant evils. In that scenario, the only way that someone could have a significant impact on the world around her is by having a significant *positive* impact; the possibility of having a significant *negative* impact on the world would be out of the question. In these circumstances, the motives for committing significant goods would be easily warped by temptations to merely display *some* exercise of significant power, rather than being marked by a *specific* desire to have a positive social impact. Consider by comparison wealthy benefactors who seek to have their family name memorialized on the buildings they help fund. We are often suspicious of their motives because it seems that they are just as, if not more, interested in having their name memorialized on an important building, as they are in the moral value of the services the building might provide. And, while we still might give them some credit if we know they *could* have chosen to use their funds for more nefarious ends, we would likely retract even that credit to the extent that it turned out God had intervened to make such more problematic ends off limits to them. In a world where God only allowed us to exercise significant power for God's pre-approved, positive ends, we would all be in the position of that kind of 'benefactor', and our motives to pursue significant goods would—for good reason—be similarly subject to suspicion.

This worry about moral corruption is especially significant if a Kantian framework for ethics is correct. While Sterba does not explicitly rely on that framework, Kantian premises provide perhaps the most natural support for the Pauline Principle that sits at the heart of his argument, and Sterba sometimes makes reference to Kantian ideas in his discussion.[12] As a result, a Kantian complaint about his argument may be particularly concerning. Kantians hold that the goodwill—the will motivated to act on the moral law for its own sake and not for any extrinsic rewards—is the only thing with unconditional, intrinsic moral worth. A world full of acts that happen to produce good consequences, but in which the people committing those acts are not acting on moral motivations, would thus (on this view) lack any significant moral value. God's decision to deprive human beings of the ability to commit *significant* evils would of course not totally deprive them of a capacity to act on moral motivations, and so such a world would not (by Kantian standards) be absent of all moral value. Nevertheless, that value would be significantly undercut by the fact that choices to exercise *significant* moral agency (by way of pursuing significant moral goods) would be significantly more prone to corruption and the significant expression of goodwill correspondingly diminished.

This worry about undercutting the operation of the goodwill is further amplified when we add on to God's interference with outer, efficacious choices for significant evils the associated constraints that such interference imposes on our inner exercise of significant, free moral agency. If the argument from the last section is correct, then—if someone decided to use her power to have a significant impact on others—that decision could never involve a free choice *between* bringing about a significant good or bringing about an alternative, significant evil. Thus, our decisions to commit significant goods would not only be liable to *becoming* corrupted over time (as was just suggested); those decisions would also *from the start* intuitively fail to have the kind of significant moral worth that genuine choices *between* significant goods and significant evils would have. Because a meaningful option to choose to commit a significant evil was removed, a person's choice to commit a significant good would no longer express her willingness or desire to *prioritize* the significant good over the correspondingly significant evil she could commit. Thus, by depriving us of a capacity for significant, inner free choice, God would once again shrink our corresponding capacities to exercise the goodwill in significant ways. The robust exercise of the one thing that Kantians take to be most critically important to realizing moral value would be further undermined.

It is difficult to say for certain what victims of significant evils would say about the above costs that would be associated with preventing their suffering. Perhaps if given immeasurable compensating, heavenly comforts and given an understanding of the nature of the trade-off God is faced with, they would themselves accept as reasonable God's choice to preserve significant, free and effective choice at the cost of allowing significant evils. However, those of us who have not undergone significant evils ourselves are not well-placed to understand the suffering of those who have been subject to those evils or (as a result) to try to evaluate how that cost of suffering 'stacks up' against the alternative of protecting significant, free and effective choice. It seems that, from our perspective, we should thus allow space for the epistemic possibility that such victims of significant evils would not consent to their suffering.

While our limited epistemic capacities may force us to remain in the dark about whether victims of significant evils will consent to their suffering and (so) whether the evidential problem of evil can be adequately addressed, God would not necessarily face this limitation. Traditional conceptions of God hold that God can foresee the future, and Molinists in particular hold that God has "middle knowledge" with respect to what individuals *would* do *if* faced with certain circumstances and possibilities for choice.[13] If Molinism is true, then God can foresee which potential victims of significant evils would (one day) come to see God's trade-off in favor of significant, free and effective choice as reasonable and—in combination with relevant heavenly comforts—thus consent to their suffering. Moreover, if God can foresee that a certain set of victims *would* consent, then God could ensure that only those victims are subject to significant evil. God could create human

beings with significant, free and effective choice while respecting the rights of victims and (so) not violating the Pauline Principle.[14]

## 6. Conclusions

Sterba argues that God would not deprive us of any morally critical form of free will if God merely blocked the final consequences of our choices for significant evil—and he might have been right if those final consequences could be blocked *without* interfering with deeper forms of free, inner moral choice or associated, robust capacities to exercise the goodwill. However, given the close connections between (outer) efficacious choice, (inner) free choice, and associated exercises of goodwill, God's blockage of the final consequences of our choices for significant evil takes on a greater significance. God would not simply be engaging in contingent interventions to prevent the effects of significant evils, but would also be shrinking our more general capacities for significant, free and effective choice and—in so doing—limiting the scope of our exercise of goodwill.

Some might try to argue that in divesting us of critical elements of significant free, inner moral agency, all for the sake of producing better consequences later, God would violate the same Pauline Principle Sterba initially appealed to in order to mount his argument. Additionally, Sterba sometimes appeals to the idea that God should protect just those freedoms that the just state would, and one might argue that in effectively 'putting out of mind' the possibility to consider and act on intentions to commit certain significant evils, God would obstruct a kind of liberty of thought or conscience with which (it is normally thought) a just state should *not* interfere. A full analysis of these further arguments, however, would require more space than is available here.

Setting aside the question of whether God would in some way wrong us by giving us the kind of limp and lopsided moral agency I have described above, the arguments I have reviewed highlight the significant moral costs of doing so. If victims of significant evils themselves regard those moral costs as prohibitive, and can—especially in light of other, compensating heavenly comforts—consent to their suffering, then God could permit significant evils even while retaining God's goodness. Additionally, as I pointed out in Section 3 (echoing the work of Scott Coley), there may, for all we know, be *other*, additional trade-offs that God must make that make God reasonable in permitting significant evils and that even victims could accept as reasonable grounds for such a permission. Still, absent knowledge of these trade-offs and of victims' attitudes about them, we cannot draw any definite conclusions. Even if a good God is logically possible, the evidential argument from evil remains unsettled.

**Funding:** This research received no external funding.

**Conflicts of Interest:** The author declares no conflict of interest.

## Notes

[1]  See Chapter 2, "There is No Free Will Defense".

[2]  For further discussion of the role of Kantian ideas in Sterba's argument, see Sections 2 and 5.

[3]  As I discuss further in the next section, Sterba does give these arguments *some* consideration (Sterba 2019, pp. 73–76).

[4]  Sterba talks interchangeably about retroactive consent and 'reasonable acceptability'. For further discussion of contractarian ideas see, e.g., (Rawls 1999; Sayre-McCord 2013; Scanlon 2000).

[5]  For other discussions of retroactive consent outside of the standard literature on contractualism, see (Carter 1977; Chang 2020; Dworkin 1972; Gersen and Suk 2017).

[6]  For classic discussions, see (Davis 1984; Harman 1997). For more recent defenses, see (Broome 2009; Ross 2009; Wallace 2001).

[7]  ibid., p. 388. For related arguments, see (Adams 1995; Mele 2022).

[8]  One might argue that God could avoid this consequence by deceiving wrongdoers and making them think that acts of significant evil had been successful even when they were not. However, traditional conceptions of a perfect God would plausibly rule out this possibility. God's goodness would intrinsically and necessarily prevent God from engaging in such deception. For further discussion of this idea, see (Murray 2008, p. 138).

[9]  This is just one among several interpretations of the idea that God exists necessarily.

10    For an introduction, see (Lewis 1970).

11    For an overview, see (Adams and Aizawa 2021).

12    See discussion of Quinn (1989) in Section 2, as well as Sterba (2019), pp. 76 and 108. Sterba also frequently frames his approach as a deontological approach, and Kantianism is perhaps the best known and most well-defended form of deontological ethics.

13    Prominent defenses of Molinism include (Dekker 2000; Flint 2018). For a survey of recent work on Molinism, see (Perszyk 2013).

14    By contrast to Molinists, open theists hold that God cannot foresee what free choices human agents will make (Hasker 2008). If open theism is true, then God cannot foresee whether victims of significant evils will consent to their suffering. Thus, God takes a significant moral risk in allowing significant evils. Still, knowing that God could not have foreseen any decision to consent (or not), victims of significant evils may still come to the conclusion that God was reasonable in taking that risk—i.e., in making a trade-off in favor significant, free and effective choice—given the cost of the alternative and given God's power to also provide immeasurably good heavenly comforts. If they judge God's risk to have been reasonable, that may provide its own grounds for consent.

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
