# Peer review of "In Answer to the Pauline Principle: Consent, Logical Constraints, and Free Will"

_religions, doi:10.3390/rel14010028_

Round 1

Reviewer 1 Report

In his 2019 book, James Sterba argues that there is a logical contradiction between the existence of God and the amount of the evil in the world, specifically, the existence of horrendous evil. Central to Sterba’s argument is the “Pauline Principle,” the idea that an agent is not morally permitted in allowing evil to befall a person in order to bring about some greater good.

However, Sterba recognizes that his Pauline Principle is not true in general. It admits many exceptions. One exception is when a person consents to undergo the evil. To take an obvious example, a person might consent to allow a doctor to put them under surgery, so long as the surgery brings about a better health outcome. Nonetheless, Sterba argues that people in the world do not generally consent to horrendous evil so this exception does not undermine his logical argument.

In this paper, the Author argues that Sterba is wrong—it is logically possible that people do consent to undergo horrendous suffering. Thus, given this exception to the Pauline Principle, there is no logical contradiction between Theism and the existence of horrendous evil. In response to the objection that people do not consent in this life, the Author invokes skeptical theism, arguing that they may consent in the next life and that we simply do not have access to such facts yet.

I found the paper to be clearly written and easy to follow. The discussion of skeptical theism could certainly be expanded beyond a discussion of Coley’s 2021 piece, but is adequate for what it is. I found the discussion of “inner free choice” and “outer effective choice” in section 4 to be insightful.

Nonetheless, I think that there are two place where the paper needs improvement:

·       Retroactive consent. The author seems to assume that people can engage in retroactive consent—that people could, in the afterlife, agree to have undergone certain experience during their live. I think the author should add a little more discussion about this type of consent. Speaking personally, it is a little odd to think a person can consent to something that has already happened. It seems more natural to think that they agree not to sanction the person who caused the harm. (Can I consent to my parent’s divorce? I don’t think so. But I could intend to not let it harm my relationship with each parent.). But I am not sure if my intuitions are representative. Regardless, additional discussion of this type of consent would be good so readers have a fuller sense of what the author is discussing.

·       Molinism. The author claims that people in the afterlife might consent to having undergone horrendous evil. But surely some people might not consent to that too. There’s no logical impossibility in being refusing such treatment by God. Thus, to avoid such a case, it seems that God would have to know which persons would consent to horrendous evil after the fact, even though they haven’t yet. But, then, it looks like the author’s account presupposes that God has a kind of “middle knowledge”—God knows which people would consent to horrendous evil in the afterlife and uses that information to determine which people may or may not undergone horrendous evil. I’m not sure that this line of reasoning is correct; that is, I’m not sure that the author’s reasoning does assume Molinism. (Nor am I objecting if it does—Molinism is a popular, if controversial, position.) However, I think many readers will have this kind of thought and so it would be good for author to discuss it.

Author Response

Dear Reviewer,

Thank you for your constructive comments. I have attached a letter with a summary of changes made in response to your comments.

Best,

Dr. Coetsee

Reviewer 2 Report

The paper “In Answer to the Pauline Principle: Consent, Logical Constraints, and Free Will” aims to present and critically discuss some main steps in Sterba’s recent work on the problem of evil (2019). The paper is organized into 6 sections. Aside from the introduction and the conclusion, respectively sections 1 and 6, four other sections work in two main stages. Sections 2 and 3 aim to defend a consent-based possibility that can excuse permitting evils, proposed as a response to the logical problem of evil. Sections 4 and 5 move toward some other challenges around freedom defense, God’s intervention, and remarks about the evidential problem of evil. I believe that the paper is well-organized and clear, leading the readers smoothly through the arguments. This is a good addition to the critical discussions of Sterba’s work, and the problem of evil more generally.

The only comment about the content I have is about section 5. I do not see how it is essential to the narrative of the paper and am not sure if it makes the case presented stronger or instead invites more questions. Plus, in this section, some assumptions are ascribed to Sterba about goodwill, based on his “Kantian framework”. I am not sure if this is a viable move, as he at the same time aims to keep the ethical theory at work in the arguments liberal enough. To ascribe such commitments there need to be clear textual evidence. I imagine that those passages in this section relevant to the previous section can be moved there, and the paper would be fine with one less section. However, I say this mostly as feedback from a reader, not a required revision.

And finally a couple of minor issues. First, one suggestion would be to mention the two stages of discussion in the abstract, the logical problem of evil and working toward addressing the evidential argument. Currently, the later sections of the paper are not reflected in the abstract, while the issues about God’s interventions might be interesting in themselves to some readers. Second, the numbering of sections (1 to 6) is not the same as mentioned in the text, where we are told about sections I-III (as mentioned in the abstract, lines 88-89, or the beginning paragraph of sec. 5.). There are also some typos, such as “playpen” and “play pen” both being used, and a seemingly to-be-removed word in “… for it view …” at line 421.

Author Response

Dear Reviewer,

Thank you for your constructive comments. I have attached a document that describes changes I have made in response to your comments.

Dr. Coetsee
